# Predatory Odor Exposure as a Potential Paradigm for Studying Emotional Modulation of Memory Consolidation—The Role of the Noradrenergic Transmission in the Basolateral Amygdala

**DOI:** 10.3390/ijms25126576

**Published:** 2024-06-14

**Authors:** Bogomil Peshev, Petya Ivanova, Desislava Krushovlieva, Lidia Kortenska, Dimitrinka Atanasova, Pavel Rashev, Nikolai Lazarov, Jana Tchekalarova

**Affiliations:** 1Institute of Neurobiology, Bulgarian Academy of Sciences, 1113 Sofia, Bulgaria; bogpeshev@gmail.com (B.P.); Ivanova.petya91@gmail.com (P.I.); daisyveko@gmail.com (D.K.); lidiakortenska@gmail.com (L.K.); didiatanasova7@gmail.com (D.A.); 2Department of Anatomy, Faculty of Medicine, Trakia University, 6003 Stara Zagora, Bulgaria; 3Institute of Biology and Immunology of Reproduction, Bulgarian Academy of Sciences, 1113 Sofia, Bulgaria; pavel_rashev@abv.bg; 4Department of Anatomy and Histology, Medical University of Sofia, 1431 Sofia, Bulgaria; nlazarov@medfac.mu-sofia.bg

**Keywords:** basolateral amygdala, noradrenaline, memory consolidation, object location task, predator odor exposure, pCREB, Arc

## Abstract

The pivotal role of the basolateral amygdala (BLA) in the emotional modulation of hippocampal plasticity and memory consolidation is well-established. Specifically, multiple studies have demonstrated that the activation of the noradrenergic (NA) system within the BLA governs these modulatory effects. However, most current evidence has been obtained by direct infusion of synthetic NA or beta-adrenergic agonists. In the present study, we aimed to investigate the effect of endogenous NA release in the BLA, induced by a natural aversive stimulus (coyote urine), on memory consolidation for a low-arousing, hippocampal-dependent task. Our experiments combined a weak object location task (OLT) version with subsequent mild predator odor exposure (POE). To investigate the role of endogenous NA in the BLA in memory modulation, a subset of the animals (Wistar rats) was treated with the non-selective beta-blocker propranolol at the end of the behavioral procedures. Hippocampal tissue was collected 90 min after drug infusion or after the OLT test, which was performed 24 h later. We used the obtained samples to estimate the levels of phosphorylated CREB (pCREB) and activity-regulated cytoskeleton-associated protein (Arc)—two molecular markers of experience-dependent changes in neuronal activity. The result suggests that POE has the potential to become a valuable behavioral paradigm for studying the interaction between BLA and the hippocampus in memory prioritization and selectivity.

## 1. Introduction

Episodic memories require the binding of information about the spatial/environmental context (“where”), the objects/landmarks encountered within the environment (“what”), the temporal order of appearance of the stimuli within the environment (“when”) and, on most occasions, the presence of familiar and/or unknown conspecifics (“who”) [1,2]. The formation of associations between the different aspects of an event is highly-dependent on the hippocampus [1,2,3,4]. Newly formed memories are labile at first, and their long-term storage requires the synthesis of specific plasticity-related proteins (PRPs) [5]. The protein-dependent process that governs the transition of a memory from its short-term form into its stable long-term one is known as memory consolidation [5]. Importantly, hippocampal glutamatergic transmission, while indispensable for memory formation, could not alone trigger the expression of PRPs that characterizes initial memory consolidation. The catecholaminergic neurotransmitters noradrenaline (NA) and dopamine (DA) as well as the glucocorticoid hormone cortisol (corticosterone in laboratory rodents) are essential for the long-term storage of memories [6,7,8]. Emotional salience and environmental novelty are two of the main conditions capable of evoking the physiological state required for the prioritization of information in the hippocampus [9,10,11]. During an emotionally arousing and/or distinctly novel event, the locus coeruleus (LC) rearranges the firing patterns of its neurons to increase the secretion of NA and DA in the dorsal hippocampus, allowing for long-term stabilization and retrieval [12,13,14]. The behavioral equivalent of the “Synaptic tagging and capture hypothesis” [15], the “Behavioral tagging hypothesis”, is a learning and memory model that offers reliable evidence of how emotional or distinctly novel experiences could modulate the consolidation of closely occurring low-arousing or neutral events [16]. According to this hypothesis, the encoding of a weak event (memory about neutral or low-emotional behavioral tasks) closely before or after the encoding of a second, strong event (emotionally arousing behavioral task novelty) will induce structural and functional modifications in sets of synapses belonging to the cellular assembly that constitutes the trace of the memory (behavioral tag) [11,16,17]. Subsequently, the elevated levels of catecholamines (or other neuromodulators like corticosterone) triggered by the strong event can activate the expression of PRPs involved in consolidation of the weak event. Thus, the neurophysiological changes induced by a salient event could prevent forgetting of the information for closely occurring neutral stimuli, since they could be used as predictors for particular aversive or rewarding outcomes in the future.

Beside the hippocampus, emotionally charged or stressful events activate the amygdala, which is involved in encoding emotional valence and emotional learning and regulating the stress responses [18,19,20,21]. There is strong evidence that, during emotional events, the basolateral amygdala (BLA) integrates arousal and peripheral hormonal signals (elevated cortisol levels) to promote a positive modulatory effect on the processes required for initial memory consolidation [9,22,23,24,25,26]. Therefore, the BLA could also influence the prioritization of information in the hippocampus [22,27]. Most studies that have established the modulatory role of the BLA on memory consolidation combined behavioral training of the experimental animals with pharmacological manipulations, mainly direct post-encoding infusions of different pharmacological agents [9]. However, evidence has also been provided by experiments using electrophysiological and optogenetic stimulation of the BLA, either during or immediately after training [28,29,30]. Importantly, post-training stimulation of the BLA affects the consolidation of various behavioral tasks regardless of the emotional valence of the training [31,32,33,34,35,36]. These findings allow for a comparison with the protocols used by the behavioral tagging studies, since the stimulation of the BLA may mimic the effects on memory consolidation produced by a strong modulatory event. Several publications have demonstrated that intra-BLA administration of the beta-adrenergic receptor agonist clenbuterol enhances memories for different tasks and increases the expression of activity-regulated cytoskeleton-associated protein (Arc) in the dorsal hippocampus [33,34,36]. Arc is an immediate early gene, the product of which is highly important for the stabilization of the synaptic modifications inducted by learning [37,38,39]. In a series of experiments, McReynolds and colleagues (2014) [34] revealed that post-encoding infusion of clenbuterol in the BLA of rats had a positive effect on the consolidation of memories for both inhibitory avoidance (IA) and novel object recognition (NOR) tasks. This was accompanied by an increase in the synaptic expression of Arc. Remarkably, the effects of clenbuterol on memory performance and Arc expression in the NOR experiment were noticeable only in the animals that were not subjected to habituation to the test conditions prior to the training. This study demonstrates that a certain degree of arousal must be induced during training in order for the BLA to exhibit its post-encoding modulatory function.

Experiments investigating BLA-induced enhancement of memory consolidation in the hippocampus, such as those described here, have led to the proposal of the “emotional tagging hypothesis” [23]. This hypothetical account is concerned with the role of the various aspects of an emotional event—intensity, controllability, duration—in facilitating the interaction between the BLA on one side and memory-related plasticity in the hippocampus on the other. Despite the great number of studies examining the emotional modulation of memory consolidation, a behavioral protocol to study the interactions mentioned above, like those implemented by the behavioral tagging studies, is still lacking.

In light of the research reviewed above, our present experimental work intends to establish a novel behavioral protocol that could provide further evidence for the role of the BLA in memory modulation by taking advantage of the natural changes in homeostatic conditions induced by an emotionally charged event. That is to say, we aimed to demonstrate that endogenous activation of the BLA by a stimulus with innate emotional valence promotes the stabilization of the behavioral tag set by a previous weak event. To do this, we decided to couple OLT with predatory odor exposure (POE) in a novel cage two minutes after the training trial. The presence of predatory cues has been proven to evoke neuronal responses in the BLA [40]. For the purpose of our experiments, we used urine from the predatory species *Canis latrans* (coyote). Urine from predatory species is a highly aversive stimulus for rodents as it contains kairomones—natural chemical substances that serve as ecological signals informing the animal of the presence of a potential predatory threat [41,42]. Previous studies have used coyote urine as an unconditioned stimulus in a hippocampus-dependent contextual fear conditioning paradigm [43,44]. A recent publication has also provided a detailed analysis of the repertoire of defensive behaviors of rats exposed to coyote urine [45]. Considering all these factors, we hypothesized that POE might be a potent modulatory event capable of inducing a strong effect on the memory consolidation for a weaker behavioral task. Furthermore, we predicted that POE-induced memory enhancement might depend on NA transmission in the BLA. To test these ideas, two separate experiments were conducted. In Experiment #1, rats were divided into three groups according to behavioral procedures and pharmacological treatment. Animals in the control group received a bilateral infusion of the vehicle into the BLA at the completion of the sample phase of the OLT. The other two groups were subjected to OLT training and exposure to coyote urine following infusion of either vehicle or propranolol into the BLA. Rats from all the groups were sacrificed 90 min after pharmacological treatment, and both of their hippocampi were extracted to examine the outcomes of BLA manipulation on experience-dependent plasticity. Experiment #2 followed the same protocol, but in order to determine a possible retroactive enhancement of memory induced by POE, the animals from the three groups were tested 24 h later. As in the first experiment, the animals were sacrificed 90 min after the test, and both hippocampi were harvested to assess the retrieval-related changes. The selected tissue samples were used to assess the levels of the phosphorylated form of cAMP response element-binding protein (pCREB) and Arc expression by ELISA and Western blotting (WB), respectively. We selected these molecular markers on the basis of existing evidence for their experience-dependent activation and role in different memory processes [37,46,47,48,49].

## 2. Results

The experimental procedure is shown in Figure 1. The details concerning the steps in Experiment #1 and Experiment #2 are shown in Section 4.

### 2.1. Experiment #1

#### 2.1.1. Post-Training Propranolol Infusion Suppresses the Effect of the Exposure to a Natural Aversive Stimulus (Coyote Urine) on Hippocampal pCREB Expression

To investigate the role of endogenous NA released from the BLA as a result of POE on the expression of plasticity marker (pCREB) in the hippocampus, the rats were trained in an OLT, which involved the exploration of two identical stimulus objects located in different positions in the sample. Immediately after the sample session, each rat from the two stressed groups was exposed to a stressful stimulus (coyote urine) for 4 min. The behaviors expressed by the experimental animals during the 4 min POE session clearly manifested the aversive nature of the selected predator odor. Through video observation, we outlined three major behavioral responses elicited by coyote urine. The animals spent more time at the end of the cage facing the Petri dish, a clear sign of avoidance behavior. Many of the rats self-groomed, which we recognized as a sign of heightened emotional arousal. Finally, some animals showed risk assessment behavior by stretching their bodies and attempting to sniff from a distance while facing the stressor. The matched control group (C + veh) was not placed in the box containing coyote urine. Immediately thereafter, the rats were infused with a vehicle into the BLA ((C + veh) and (POE + veh) or propanolol (POE + prop)). At 90 min after the sample test, the hippocampus was isolated from all groups. One-way ANOVA revealed a significant difference between groups [F2,20 = 3.809, *p* = 0.0418] for the expression of activated CREB (pCREB) in the hippocampus at the sample phase. The post hoc test showed that pCREB expression in the hippocampus was significantly increased in the POE + veh group compared to the C + veh group (*p* = 0.0011) (Figure 1). At the same time, post-training infusion of propranolol after training suppressed the expression of this transcription factor in the hippocampus (*p* = 0.0013 compared to the POE + veh group).

#### 2.1.2. The Arc Expression in the Hippocampus Showed a Tendency for Elevation as a Sequence of Post-Training Exposure to a Natural Aversive Stimulus (Coyote Urine)

To ascertain whether the Arc protein expression in the hippocampus is involved in the effect of endogenous NA released from the BLA on the consolidation process, the three groups (C + veh, POE + veh and POE + prop) were decapitated 90 min after the sample test. The Arc expression was explored with the Western blot (WB) test (Appendix A). A tendency for increased Arc expression in the hippocampus was detected in the POE + veh group compared to the C + veh group in homogenates tested by the WB test (Figure 2).

### 2.2. Experiment #2

#### 2.2.1. Acute Post-Training Exposure to a Natural Aversive Stimulus (Coyote Urine) Did Not Affect the Performance of Rats on the Object Location Task

The procedure for infusing vehicle/propranolol into the BLA of rats in Experiment #2 was the same as in Experiment #1. Twenty-four hours after training, each rat from the three groups was re-exposed to two identical objects placed in a different location (Figure 3). One-way ANOVA analysis revealed no significant difference between groups [F2,25 = 1.402, *p* > 0.05]. Although there was a tendency for impaired retention in the stress + prop group, the results suggest a lack of memory-enhancing effect of acute stress-induced release of NA from the BLA.

#### 2.2.2. Acute Post-Training Exposure to a Natural Aversive Stimulus (Coyote Urine) and Propranolol Infusion Did Not Change the pCREB Expression in the Hippocampus in the Retention Phase

No significant difference was found between the three groups after the test session for the pCREB expression in the hippocampus [F2,24 = 0.824, *p* > 0.05] (Figure 4), suggesting a lack of effect of endogenous NA released from the BLA after acute stress on this marker of plasticity.

#### 2.2.3. Acute Post-Training Exposure to Predator Odor and Propranolol Infusion Did Not Change the Arc Expression in the Hippocampus in the Retention Phase

There was no significant difference between the three groups (C + veh, POE + veh and POE + prop) in Arc expression in the hippocampus, measured by the WB test, in the retention phase (Figure 5).

## 3. Discussion

The experiments presented here were designed to demonstrate that NA activation of the BLA during emotionally arousing situations is critical for any retrograde enhancement effect on memory consolidation. As our protocol relied on exposing animals to a stimulus with an innate negative valence to induce emotional arousal rather than a pharmacological intervention, we aimed to provide direct evidence for the emotional aspects of the behavioral tagging. Our study is conceptually close to that of da Cunha and colleagues (2019) [50]. They paired weak OLT (4 min sample session) with an elevated platform (EP) session 1 h after the initial training and found that the induced state of acute stress was able to rescue the memory for the OLT from forgetting. The observed effects were dependent on elevated levels of corticosterone in the dorsal hippocampus, since infusion of both mineralocorticoid (MRs) and glucocorticoid receptor (GRs) antagonists into the CA1 field disrupted the effects induced by the stressful experience. Although this was not the focus of their experiment, the results obtained by the researchers regarding the contribution of hippocampal MRs and GRs to memory enhancement could be interpreted in terms of the BLA-dependent process. As mentioned above, McReynolds at al. (2010) [32] showed that the beneficial effect of a systemic corticosterone injection on memory could be prevented by infusion of propranolol into the BLA. These observations further motivated us to investigate how endogenous NA release in the BLA during emotionally arousing situations influences the dynamics of memory formation in the hippocampus.

The main finding of our study is that the animals that were exposed to the predator odor and infused with vehicle immediately after the end of the OLT sample session showed a significant upregulation in their pCREB levels and a trend towards increased Arc expression in the hippocampus compared to those infused with propranolol. Most importantly, propranolol, infused into the BLA after completion of the sample session and POE, reduced the expression of this transcription factor, suggesting the role of endogenous NA released from the BLA. Thus, the significant increase in pCREB levels and the tendency for higher Arc expression in the rats from the POE/veh group compared to the control group highlights the influence of the emotional event on hippocampal plasticity. A previous study by Kabitzke, Silva and Wiedenmayer (2011) [51] reported that using cat odor as an unconditioned stimulus in a contextual fear conditioning paradigm significantly increased pCREB levels in both the dorsal and ventral hippocampus of juvenile rats. However, when an intraperitoneal injection of propranolol was administered after the exposure to the predatory cue, this experience-dependent effect was not observed, and the animals showed no conditioned response when they were re-exposed to the training context 24 h later, suggesting that the antagonist may have blocked the activation of CREB by NA and suppressed signaling mechanisms related to consolidation. In contrast, post-training systemic injection of the non-selective beta-adrenergic blocker nadolol, which cannot cross the blood–brain barrier, had no effect on pCREB levels or the behavioral responses studied, suggesting that the effect on memory consolidation was facilitated by the endogenous release of NA in the brain. Our findings regarding pCREB extend the evidence described above and show that the effects of predatory cues on hippocampus-dependent memory and memory-induced behavioral responses are also influenced by the activation of the noradrenergic neurotransmission in the BLA. Importantly, in the experiment of Kabitzke, Silva and Wiedenmayer (2011) [51], the hippocampal tissue was collected 60 min after odor exposure. In our study, both hippocampi were extracted 90 min after the completion of all experimental procedures. This timepoint was chosen because we wanted to establish the possible existence of a prolonged window for CREB activation. Since CREB is one of the main transcription factors involved in memory consolidation [47,48], a prolonged activation of CREB can be interpreted as an indication of increased transcriptional activity of the genes responsible for PRPs. With regard to Arc, our results showed a trend towards increased expression of the protein in the animals from the POE/veh group. This further emphasizes the role of BLA in modulating memory-related plasticity in the hippocampus during emotionally charged events. The studies providing evidence for BLA-modulated Arc expression have used synaptoneurosome preparation to determine synaptic levels of the protein in the dorsal hippocampus [32,34,36]. In addition, these studies have collected the samples at different timepoints and showed that the peak concentration can be approximately 1 h after training [32,36]. In our experiment, we homogenized the whole hippocampus. This, together with the timepoint of tissue collection, may have limited our ability to obtain more detailed results. Nevertheless, we still were able to demonstrate the presence of an emotional arousal-dependent effect on Arc expression regulated by the BLA.

Surprisingly, in the second experiment, the POE/veh group did not express stronger discriminative behavior during the test session; this response corresponded to the lack of difference in both the CREB and Arc expressions between the three groups. Our results are in line with Moncada and Viola (2006), who report [49] that a familiar spatial environment can reduce the phosphorylation of the protein. With regard to the behavioral procedures, in both experiments, the animals were habituated to the experimental context for 10 min, one day prior to the sample session of OLT and POE. This was performed in order to limit the presence of possible anxiety-related and neophobic behaviors during the training. However, as it was already mentioned, McReynolds et al. [42] reported that habituation may neutralize the influence of post-encoding BLA activation on memory consolidation. In their paper, McReynolds et al. (2014) [34] suggest that there may be a requirement for parallel arousal-induced activation of both the BLA and the efferent memory-related structures for the modulatory effect to occur and promote greater synaptic expression of Arc. Thus, by familiarizing the animals with the experimental context, the habituation session of the current experiments may have limited the modulatory influences of the BLA by reducing novelty-induced activation of the dorsal hippocampus. Notably, in a study by Ballarini and colleagues (2009) [52]—which coupled a weak spatial object recognition task with OF—animals were subjected to two habituation sessions, but the presence of an enhancing effect on memory consolidation was still observed. This suggests that the modulatory events characterized either by environmental novelty or the presence of emotionally arousing stimuli may promote memory consolidation by activating different neural circuits. Confirmation or rejection of this notion requires more behavioral tagging studies that include emotionally valenced stimuli in their modulatory events. Another factor that may have influenced the lack of noticeable modulatory effect on memory consolidation in our second experiment is the temporal window between the weak and the modulatory events. The rats underwent POE only two minutes after the end of the OLT sample session. This interval was chosen because we wanted to design a more naturalistic behavioral paradigm. In natural environmental conditions, emotional stimuli may appear surprisingly and influence the consolidation of the memories for the most proximal neutral stimuli or contextual features that could be used to optimize future memory-based predictions. Most behavioral tagging experiments that reported positive modulation on memory consolidation have exposed the animals to the strong event at different timepoints—ranging from 15 min to 1 h before or after training [18,20,22]. Importantly, da Cunha and colleagues (2022) [53] found that acute stress (EP) experienced immediately after wIA did not impair the formation of memory for the task and facilitated its long-term stabilization. However, given the different nature of the emotionally arousing events selected in ours and theirs (POE vs. EP), we cannot currently rule out the possibility that the animals in our experiment may have formed a strong association between the two events that could have influenced their performance during the test session of OLT. When re-exposed to the experimental context the following day, the animals may have expected to encounter the aversive stimuli once again, which may explain why we did not observe the memory-related behaviors we expected. In a recent review by Sazma, Shields and Yonelinas (2019) [54], which examined the effects of post-encoding stress in animal and human studies, the authors have expressed similar concerns about the possible negative effect of acute stress on subsequent memory retrieval. Considering the fact that the POE was only performed for 4 min, we doubt that the intensity of our modulatory event could have negatively impacted the consolidation of the memory for OLT. Detailed studies on how different physiological parameters of the emotional response to different predatory odors might influence the cellular and molecular basis of memory processes are still lacking. Nevertheless, a recent report by Migliaro and colleagues (2019) [55] provides some insights into the observed behavioral changes depending on the duration of the exposure. Using various tests to assess anxiety-like and depressive-like behavior 24 h after exposure of Wistar rats to bobcat urine, they found that animals exposed for 10 or 20 min showed a clear change in their behavioral profile, while those exposed to the predatory cue for 3 min showed no profound alterations. Taking all these different possibilities into account, we must emphasize that the results of our first experiment on pCREB and Arc expression directly argues against the likelihood of our modulatory event being too mild to produce experience-dependent neural activity in the hippocampus.

In order to further validate the physiological effects of the selected aversive stimulus (coyote urine) and to extend the protocol proposed in the current article, we may need to add a group of experimental animals exposed to a neutral odor or an empty Petri dish. Another interesting approach will be to habituate the animals to the cage in which the POE is performed in order to eliminate the effects of spatial novelty. In this way, and by taking advantage of methods such as immunohistochemistry, we will be able to obtain a more detailed view of the expression of Arc induced by the predatory odor and the distribution of the protein in the BLA, the hippocampus and other memory-related cortical areas, such as the entorhinal and retrosplenial cortices.

## 4. Materials and Methods

### 4.1. Animals

Sixty-day-old male Wistar rats (300–350 g) were purchased from the vivarium of the Institute of Neurobiology, BAS. Upon arrival, they were acclimated for one week prior to surgery. The rats were maintained under standard environmental conditions (22–23 °C, 12/12 h light/dark cycle with light on at 7 a.m.) in groups of 3–4 in Plexiglas cages with access to laboratory chow and tap water ad libitum.

The experiments were performed in accordance with the Council Directive 2010/63/EC on animal experiments and approved by the Bulgarian Food Safety Agency (research project: N^◦^349).

### 4.2. Experimental Design

The experiments were performed in two separate protocols. The rats were randomly (blindly) allocated to Experiment 1 (pCREB and Arc expression in the sampling phase) and Experiment 2 (retention test, pCREB and Arc expression). The rats in one of three different conditions (n = 6–8) were as follows: Group I (C + veh), Group II (POE + veh) and Group III (POE + prop). The rats were anaesthetized with ketamine (80 mg/kg, i.p.) and xylazine (20 mg/kg, s.c.) and implanted bilaterally in the BLA by stereotaxic surgery with two cannulas according to the atlas of Paxinos and Watson [56] at the following coordinates: AP = −2.5 L = ±4.8; H = 6.5 (Appendix A). One week after recovery, the animals from the three groups were habituated for 10 min in the open field used for the memory task. Twenty-four hours later, the rats were trained in the OLT (a probe phase). Two minutes after training, each rat from Group II and Group III was exposed to a predator odor (coyote urine) for 4 min. Group I was not exposed to the box containing a predator odor. The predator odor exposure was conducted in the same room as the OLT, two minutes after the end of the sample session of the memory task. During this short interval, a Petri dish containing a gauze previously soaked in 7 mL of coyote urine (Main Outdoor Solutions) was randomly placed in one of the four corners of a transparent cage (50 cm × 50 cm × 80 cm). The experimental animals were not previously accustomed to the cage. Before the beginning of the behavioral procedure, the Petri dish was kept in a sealed plastic box outside the experimental room to avoid the spread of the scent. The rat was then placed in the cage for 4 min. A video camera placed above the cage was used during each trial to observe the emotional behaviors expressed by the rats. Since the animals were not habituated to the cage and spatial novelty could be a strong modulator of memory consolidation, the current protocol did not include a separate group of rats placed in the same environment but with an empty Petri dish.

The rats were injected with vehicle (Groups I and II) or propranolol (0.75 µg) using a 30-gauge needle extended 2 mm beyond the cannula. The injection needle was connected to a 1 µL Hamilton syringe via a silicone tube. The dose of the beta-adrenergic receptor antagonist used was based on the previous report of Segev and Ramot (2012) [57]. A volume of 0.5 µL vehicle (saline) or propranolol (0.5 µL/60 s) was infused. The injection needle was left in place for a further 60 s, after which the rat was returned to its home cage.

### 4.3. Object Location Task

The object location task was conducted in an OF box (50 cm × 50 cm × 50 cm) with grey walls. The procedure consisted of three separate trials: familiarization, probe (training) and retention. A video camera placed above the box was used in each trial, and the video recordings were analyzed offline. During habituation, each rat was placed in the OF without an object for 10 min. Twenty-four hours later, the sample trial was performed, and each tested rat was exposed to two identical plastic objects (A1 and A2) for 5 min, positioned as shown in (Figure 1). The retention trial was performed 24 h later, and the tested subject was placed in the same OF box, where one of the objects remained in the same position as in the sample trial (A1) and the other object (A2) was placed in a novel position (Nov) for 3 min. The exploration time (sec), defined as the nose approaching the object in the test trial, was scored. The discrimination index (DI) for each object was calculated as follows: DI = Nov/A1 + Nov. The OF box and objects were thoroughly cleaned with 70% alcohol (*v*/*v*) after each trial. The animals were sacrificed by guillotine, and left and right hippocampal tissue was isolated, immediately frozen in liquid nitrogen and stored in a refrigerator at −20 °C. Hippocampal tissue was collected 90 min after drug infusion or after the OLT test, which was performed 24 h later. This timepoint was selected based on an article by Lonergan et al. (2010) [58].

### 4.4. Enzyme-Linked Immunosorbent Assay (ELISA) Test

Hippocampal samples were weighed and homogenized with HEPES buffer (20 mM HEPES; 1 mM EGTA; 210 mM mannitol; 70 mM sucrose; pH 7.2) and protease inhibitor cocktail (100 mM PMSF, 100 mM NaF, 35 mM EDTA). The homogenates were centrifuged at 10,000× *g* for 5 min at 4 °C. The supernatants were used for the determination of total protein by the Bradford method. Thereafter, two ELISA tests were performed to determine the protein expression of Arc and pCREB. The ELISA assay was performed according to the manufacturer’s instructions for the specific kits (ELISA kit for Activity Regulated Cytoskeleton Associated Protein (ARC), SEJ620Ra; Rat phospho-cAMP response element binding protein (p-CREB), SL1344Ra).

### 4.5. Western Blot Test

Homogenized hippocampal tissue samples and protease inhibitor cocktail were mixed 1:1 with glycerol loading buffer, loaded on four 12% SDS acrylamide gels and separated by electrophoresis (Biorad Mini-PROTEAN Tetra Vertical Electrophoresis Cell) for 15 min on 90 V and 1 h at 120 V. The proteins were transferred to nitrocellulose membranes by semi-dry transfer (Biorad Trans-Blot^®^ SD Semi-Dry Transfer Cell) 10 V for 30 min. Non-specific binding was blocked by incubating overnight with 5% (*w*/*v*) BSA in TBS. The membranes were then incubated for two hours at 37 °C with anti-Arc primary antibodies (Affinity Biosciences, Hong Kong, China) diluted 1:500 in TBS + 5% BSA buffer. The membranes were washed with TTBS and incubated with HRP-conjugated goat anti-rabbit IgG (Elabscience, E-AB-1003, Houston, TX, USA) for one hour at room temperature. After washing, the bands were detected using chemiluminescent dye (Abcam ab134640, Kambridge, UK) using a chemiluminescent blot scanner (C-DiGit^®^ Blot Scanner for Chemiluminescent Western Blots) by scanning for 6 min. Weight electrophoresis pre-stained protein marker (Elabscience E-BC-R273) was run on the same slab gels to determine the relative molecular weight of the proteins.

### 4.6. Statistical Analysis

The experimental data were presented as mean ± S.E.M. The parametric statistical test was used in case of normality data distribution (one-way ANOVA followed by Tukey’s test), while the Kruskal–Wallis on ranks followed by the Mann–Whitney U test was applied for non-parametric data. SigmaStat^®^ (version 11.0., San Jose, CA, USA) 362 and GraphPad Prism 6 software were used for the statistical analyses. The significance level was set at *p* < 0.05.

## 5. Conclusions

Taking into account the result of Experiment #1, we are confident that, with more detailed adjustments, POE could become an even more suitable strategy for investigating the complex neurobiological systems behind emotional modulation of memory processes in laboratory rodents. In the future, it will be interesting to couple POE with other weak behavior protocols that have been extensively used to study hippocampal-dependent memory processes and behavioral tagging (contextual fear conditioning, inhibitory avoidance). As it was outlined in the discussion section, the temporal interval between the weak behavioral task and POE needs to be carefully balanced to exclude any negative effects of the aversive stimulus on learning. In addition, the use of methods such as immunohistochemistry and in situ hybridization will allow for us to better understand the interaction between POE and the learning task it modulates at the circuit level. By activating the neural circuits involved in threat detection and defensive behaviors, predatory odors induce heightened emotional arousal. Studies have shown that the BLA is one of the main structures responsible for regulating the unconditioned response towards predatory odors. However, to our knowledge, no previous experiment has attempted to examine possible modulatory influences of the emotional state induced by predatory cues on the consolidation of memories for events. Thus, here, we attempted to bridge two distinct and independently developing research lines. The behavioral tagging hypothesis offers exciting opportunities to study the effects of different internal states and environmental conditions on memory formation. The result of such studies may provide valuable insights into the search for novel therapeutic strategies targeting the pathological alterations of episodic memory associated with Alzheimer’s disease and affective disorders. Emotional arousal is a strong modulator of hippocampal plasticity, and investigating how its effects on memory are regulated by the noradrenergic neurotransmission in the BLA will provide a more in-depth understanding of the mechanism for information prioritization.

## Data Availability

All data generated or analyzed during this study are included in this article. Further inquiries can be directed to the corresponding author.

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
