# Peer review of "Predatory Odor Exposure as a Potential Paradigm for Studying Emotional Modulation of Memory Consolidation—The Role of the Noradrenergic Transmission in the Basolateral Amygdala"

_ijms, 2024, doi:10.3390/ijms25126576_

Round 1
Reviewer 1 Report (Previous Reviewer 2)
Comments and Suggestions for Authors
The authors fully addressed all concerns raised by me. Congratulations to the authors on an interesting study.
Comments on the Quality of English LanguageEnglish language is fine but minor text editing may be required.
Author Response
We appreciate the Reviewer's comment!

Reviewer 2 Report (New Reviewer)
Comments and Suggestions for Authors
The abstract provides a clear and concise summary of the research focus, methods, and key findings. It effectively outlines the main aim of the study, which is to investigate the effect of endogenous noradrenergic (NA) release in the basolateral amygdala (BLA) on memory consolidation using a natural aversive stimulus, specifically coyote urine. The use of a weak object location task (OLT) combined with predator odor exposure (POE) is well-explained, and the role of propranolol in the study design is adequately described. The mention of phosphorylated CREB (pCREB) and activity-regulated cytoskeleton-associated protein (Arc) as molecular markers is appropriate, given their relevance to memory processes.
Key Points and Contributions
1. **Innovative Paradigm**: The study proposes an innovative paradigm by using a natural aversive stimulus (coyote urine) to induce endogenous NA release in the BLA, which is a significant advancement over previous methods that relied on synthetic NA or beta-adrenergic agonists.
2. **Behavioral and Molecular Approaches**: Combining behavioral testing (OLT) with molecular analysis (pCREB and Arc levels) is a comprehensive approach that strengthens the study's conclusions about the interaction between BLA and hippocampus in memory modulation.
3. **Potential Applications**: The findings suggest that POE could become a valuable tool for studying emotional modulation of memory, which has broad implications for understanding the mechanisms underlying memory prioritization and selectivity.
Doubts and Areas for Clarification
1. **Choice of Propranolol**: While propranolol is a well-known beta-blocker, it would be helpful to understand why it was chosen over more selective beta-adrenergic antagonists. This could impact the specificity of the findings related to the NA system.
2. **Timing of Measurements**: The rationale behind the specific timing of hippocampal tissue collection (90 minutes post-drug infusion and 24 hours post-OLT) could be elaborated. Why were these time points chosen, and how do they relate to the expected dynamics of NA signaling and memory consolidation?
3. **Control Conditions**: Detailed information on control conditions is essential. For instance, were there control groups exposed to a neutral odor instead of predator odor? This would help in isolating the effects of POE from other potential stress-related factors.
4. **Extent of Arousal**: The abstract mentions a "low-arousing hippocampal-dependent task" but does not quantify or describe how arousal levels were measured or controlled. Clarifying this point is important to understand the baseline conditions under which memory modulation was studied.
5. **Generalizability of Findings**: Given that the study was conducted on Wistar rats, it would be valuable to discuss the potential generalizability of the findings to other strains or species. Are there known differences in NA system responses that might affect the outcomes?
The article presents a novel and potentially impactful method for studying the emotional modulation of memory consolidation through endogenous NA release in the BLA. The use of a natural aversive stimulus (coyote urine) combined with a behavioral task and molecular analysis is a strong approach. However, some clarifications regarding the choice of propranolol, timing of measurements, control conditions, and the generalizability of findings would strengthen the study. Overall, the research offers promising insights and could pave the way for further investigations into the interplay between emotional arousal and memory processes.
Comments on the Quality of English LanguageIt's fine.
Author Response
Review # 2
The abstract provides a clear and concise summary of the research focus, methods, and key findings. It effectively outlines the main aim of the study, which is to investigate the effect of endogenous noradrenergic (NA) release in the basolateral amygdala (BLA) on memory consolidation using a natural aversive stimulus, specifically coyote urine. The use of a weak object location task (OLT) combined with predator odor exposure (POE) is well-explained, and the role of propranolol in the study design is adequately described. The mention of phosphorylated CREB (pCREB) and activity-regulated cytoskeleton-associated protein (Arc) as molecular markers is appropriate, given their relevance to memory processes.
Key Points and Contributions
- **Innovative Paradigm**: The study proposes an innovative paradigm by using a natural aversive stimulus (coyote urine) to induce endogenous NA release in the BLA, which is a significant advancement over previous methods that relied on synthetic NA or beta-adrenergic agonists.
- **Behavioral and Molecular Approaches**: Combining behavioral testing (OLT) with molecular analysis (pCREB and Arc levels) is a comprehensive approach that strengthens the study's conclusions about the interaction between BLA and hippocampus in memory modulation.
- **Potential Applications**: The findings suggest that POE could become a valuable tool for studying emotional modulation of memory, which has broad implications for understanding the mechanisms underlying memory prioritization and selectivity.
Doubts and Areas for Clarification
Point #1 1. **Choice of Propranolol**: While propranolol is a well-known beta-blocker, it would be helpful to understand why it was chosen over more selective beta-adrenergic antagonists. This could impact the specificity of the findings related to the NA system.
Response: The choice of propranolol treatment was based on the report by McReynolds at al. (2010), who showed that systemic corticosterone injection on memory could be prevented by infusion of propranolol into the BLA (line 237). The dose and mode of propranolol treatment were based on the report by Segev A, Ramot A, Akirav I. 2012. This citation was inserted in the text (Material and Methods section).
Point #2. **Timing of Measurements**: The rationale behind the specific timing of hippocampal tissue collection (90 minutes post-drug infusion and 24 hours post-OLT) could be elaborated. Why were these time points chosen, and how do they relate to the expected dynamics of NA signaling and memory consolidation?
Response: This time point was selected based on an article by Lonergan et al. (2010) - DOI: 10.1155/2010/139891. By selecting this time point, we were hoping to demonstrate that POE could induce a window of prolonged plasticity in the hippocampus. The obtained results regarding pCREB and Arc confirmed our expectations. However, this time point may not have been the best option after the OLT test as retrieval in a familiar environment may not be able to trigger Arc expression as powerful as in the conditions of novel or emotionally charged environment
Point #3. **Control Conditions**: Detailed information on control conditions is essential. For instance, were there control groups exposed to a neutral odor instead of predator odor? This would help in isolating the effects of POE from other potential stress-related factors.
Response: We thank you for raising this important point. In fact, we decided to omit the use of a neutral odor petri in the control group in order to eliminate any additional conditions (such as the use of a new environment - a box with a petri filled with neutral odor) that could be considered "novelty". However, the reviewer is correct that an additional group in the above conditions is necessary to assess the true effect of POE. We have mentioned this limitation of our paradigm in the Discussion section and the need to use this additional group in future implementations of this paradigm (line 355).
Point #4. **Extent of Arousal**: The abstract mentions a "low-arousing hippocampal-dependent task" but does not quantify or describe how arousal levels were measured or controlled. Clarifying this point is important to understand the baseline conditions under which memory modulation was studied.
Response: As we explained in the introduction (line 55), our paradigm is based on the "behavioral tagging hypothesis" by applying an emotionally novel experience (POE) after a low-arousing (or low-intensity) neutral event (OLT), which is expected to modulate consolidation. The low-arousing neutral event (OLT) or ORT as a behavioral tag has also been used in several other studies (Ballarini et al., 2009; Medha Kaushik et al., 2023). The OLT test, without the use of a novel experience in a critical time window, will induce short-term memory but will not be able to provoke long-term memory. The OLT is based on the spontaneous tendency of rodents, previously exposed to two identical objects, to later explore one of the objects - replaced in a novel location - for longer than the nondisplaced object. This memory test is used without conventional reinforcers (positive or negative) and is considered to be more appropriate for studying the neural mechanisms underlying cognitive performance i.e. this test is non-stressful and is widely used to test STM. The lack of stimuli with learned or innate emotional valence in this paradigm means that the arousal systems (like locus coeruleus and ventral tegmental area) are only partially activated, most likely due to the novelty of the object. Low activation of the systems that release noradrenaline and dopamine in the hippocampus can not induce protein synthesis in the sets of synapses that were active during the experience. Without de novo synthesized plasticity-related proteins the LTP can not transition from its unstable early phase to its stable late phase.
Point #5. **Generalizability of Findings**: Given that the study was conducted on Wistar rats, it would be valuable to discuss the potential generalizability of the findings to other strains or species. Are there known differences in NA system responses that might affect the outcomes?
Response: The reviewer is absolutely right that in the future this paradigm needs to be verified in other strains or species to prove it to be universal. This study was a pilot investigation that will pave the way for future investigations with other strains and will be useful to explore the interaction of NA with other modulatory systems (such as DA or glucocorticosteroids).
Point #6 The article presents a novel and potentially impactful method for studying the emotional modulation of memory consolidation through endogenous NA release in the BLA. The use of a natural aversive stimulus (coyote urine) combined with a behavioral task and molecular analysis is a strong approach. However, some clarifications regarding the choice of propranolol, timing of measurements, control conditions, and the generalizability of findings would strengthen the study. Overall, the research offers promising insights and could pave the way for further investigations into the interplay between emotional arousal and memory processes.
Response: 1) The use of the beta-adrenergic receptor antagonist propranolol is explained in the Discussion section (from line 248). The choice of propranolol was based on the report by McReynolds at al. (2010), who showed that systemic corticosterone injection on memory could be prevented by infusion of propranolol into the BLA
2) Timing of measurements. In the discussion, we explained why we used an interval of 4 min for the POE (line 342 and from line 348) (Considering that the POE was performed for only 4 min, we doubt that the intensity of our modulatory event could have negatively affected the consolidation of the memory for OLT) and (Using various tests to assess anxiety-like and depressive-like behavior 24 h after exposure of Wistar rats to bobcat urine, they found that animals exposed for 10 or 20 minutes showed a clear change in their behavioral profile, while those exposed to the predatory cue for 3 minutes showed no profound changes...).
3) Control conditions. We added a text for control conditions in the Materials and Methods with more details about the control group (line 388).
4) The generalizability of the results would strengthen the study. The answer is the same as in point #5.
We inserted also a paragraph explaining in details the procedure of POE (line 388),

Reviewer 3 Report (New Reviewer)
Comments and Suggestions for Authors
In this study, Peshev et al investigated the effect of the Coyote urine-induced noradrenaline release in the basolateral amygdala (Using NA antagonist), and on memory-related task. They did some behavioral experiments and biochemical assays like ELISA and Western blot to conclude that predator odor exposure has the potential to become a valuable behavioral paradigm for studying the crosstalk between BLA and the hippocampus in memory prioritization and selectivity. Here are my comments:
Major:
1. How did authors confirm the cannula location in BLA? Any histo or CV or thionin staining, etc experiment visualizing the cannula location should be shown in this paper.
2. Line 371-372: :A volume of 0.5 µl vehicle (saline) or propranolol (0.5 µl/60 s) was infused”. How precise is the very small amount of the drug delivery using a 5mL manual Hamilton Syringe? The field generally uses low volume syringes for this purpose. Please clarify.
3. What was the rationale behind selecting propranolol dose (0.75 µg). Was a pilot study performed for this? Please give a reference if dose was selected as per the previous literature.
4. Although authors have performed ELISA and Western Blot to convince the outcome of this study, to increase the strength of their work, they could also show some IHC or ISH results using a neuronal marker like cFos+NeuN (and may be Arc!) and could show the up- or down-regulation correlated to their behavioral paradigms.
5. What about the anxiety behavior, etc. in this study. This study lacks very commonly used behavioral assays. Since authors recorded the videos, they should also add the behavioral parameters like anxiety behavior, freezing time, rearing, etc by reanalyzing their videos for all groups .
Minor:
1. Experiment Fig. 1 and 2 on page #4 can be combined with the similar experimental Fig. 1 and 2 on page #5 for more specific experimental scheme.
2. In Fig. 3, what is the unit of Time at Y axis. If it is discrimination Index then please clarify? Please explain in Fig. Legend.
3. Re-frame sentence in line #349 (Use SI Unit of weight should be ‘g’ not ‘gr’).
4. In statistical analysis section, please provide the name of software used.
5. The predator odor induces autonomic fear responses. Did authors record any physiological output like Urine & Boli Index?
Comments on the Quality of English Language
It is ok.
Author Response
Review # 3
In this study, Peshev et al investigated the effect of the Coyote urine-induced noradrenaline release in the basolateral amygdala (Using NA antagonist), and on memory-related task. They did some behavioral experiments and biochemical assays like ELISA and Western blot to conclude that predator odor exposure has the potential to become a valuable behavioral paradigm for studying the crosstalk between BLA and the hippocampus in memory prioritization and selectivity. Here are my comments:
Point #1 Major:
- How did authors confirm the cannula location in BLA? Any histo or CV or thionin staining, etc experiment visualizing the cannula location should be shown in this paper.
Response: We verified the location of the cannula in the BLA by histological method (cresyl violet staining). These data have been included in the Supplementary Appendix.
Point #2. Line 371-372: :A volume of 0.5 µl vehicle (saline) or propranolol (0.5 µl/60 s) was infused”. How precise is the very small amount of the drug delivery using a 5mL manual Hamilton Syringe? The field generally uses low volume syringes for this purpose. Please clarify.
Response: Thank you for this comment! This was a technical error in the description! We used a 1 µl Hammilton syringe. The error has been corrected.
Point #3. What was the rationale behind selecting propranolol dose (0.75 µg). Was a pilot study performed for this? Please give a reference if dose was selected as per the previous literature.
Response: The dose and mode of propranolol treatment were based on the report by Segev A, Ramot A, Akirav I. (2012). This citation was inserted in the text (Material and Methods section). Other memory modulations studies such as the one by McReynolds et al. (2010) also take advantage of direct infusion of propranolol in the BLA in order to study effect on memory consolidation in the hippocampus
Point #4. Although authors have performed ELISA and Western Blot to convince the outcome of this study, to increase the strength of their work, they could also show some IHC or ISH results using a neuronal marker like cFos+NeuN (and may be Arc!) and could show the up- or down-regulation correlated to their behavioral paradigms.
Response: This study was our lab's first attempt to study the effects of endogenous NA release in the BLA the effect of endogenous noradrenergic (NA) release in the basolateral amygdala (BLA) on memory consolidation using a natural aversive stimulus (coyote urine). We are grateful for the reviewer's suggestion. After validating this paradigm with the present study and with some careful adjustment in the near future, we plan to use immunohistochemistry to investigate the signaling molecules involved in consolidation in different subfields of the hippocampus.
Point #5. What about the anxiety behavior, etc. in this study. This study lacks very commonly used behavioral assays. Since authors recorded the videos, they should also add the behavioral parameters like anxiety behavior, freezing time, rearing, etc by reanalyzing their videos for all groups .
Response: As we explained in the introduction (line 55), our paradigm is based on the "behavioral tagging hypothesis" by applying an emotionally novel experience (POE) after a low-intensity neutral event (OLT), which is expected to modulate consolidation. Video analysis was used to monitor the response of each rat (in groups with a POE) when was placed in the coyote urine box. In general, all rats escaped the urine-filled petri went to the opposite corner of the box. We added a sentence in Result section (line 160) describing this behavioral reaction to POE.
Minor:
Point #1. Experiment Fig. 1 and 2 on page #4 can be combined with the similar experimental Fig. 1 and 2 on page #5 for more specific experimental scheme.
Response: Corrected.
Point #2. In Fig. 3, what is the unit of Time at Y axis. If it is discrimination Index then please clarify? Please explain in Fig. Legend.
Response: In the new version of the manuscript, “time” has been deleted after DI on the ordinate. The formula for calculating of DI is given in Materials and Methods section.
Point #3. Re-frame sentence in line #349 (Use SI Unit of weight should be ‘g’ not ‘gr’).
Response: Corrected.
Point #4. In statistical analysis section, please provide the name of software used.
Response: The name of the software used was added in the new version of the manuscript.
Point #5. The predator odor induces autonomic fear responses. Did authors record any physiological output like Urine & Boli Index?
Response: As an indicator of escape behavior, we detected only the response of avoidance of urine-filled petri dishes.

Round 2
Reviewer 3 Report (New Reviewer)
Comments and Suggestions for Authors
I have no further comments and satisfied with the authors' responses.
Comments on the Quality of English LanguageIt seems ok to me.
This manuscript is a resubmission of an earlier submission. The following is a list of the peer review reports and author responses from that submission.
Round 1
Reviewer 1 Report
Comments and Suggestions for Authors
The authors aim to study the effect of endogenous NA release in the BLA induced by an aversive stimulus on memory consolidation for a neutral hippocampal-dependent task.
For that they used the behavioral paradigm object location task (OLT) combined with mild predator odor exposure (POE). At the same time the authors aimed at showing that this paradigm can be used for the study of emotional modulation of memory consolidation.
I found the title of the article very confusing and the behavioral results obtained are not really in agreement with this title.
Also, the introduction is very long. The authors do not need to give that many details in this section.
The authors predicted that memory enhancement prompt by POE may be dependent on NA transmission in BLA. However, the results of this article do not support this.
- First the results do not show that POE is capable of inducing a strong effect on the memory consolidation for a weaker behavioral task (Fig. 3)
- Additionally, if the authors would also need to measure/confirm the release of NA in the BLA during their paradigm to full state that the stress induced by POE is dependent on NA transmission in BLA.
There are some discrepancies across the manuscript in what concerns the experimental procedures between scheme 1 and the text:
- Habituation to the OF box was mentioned to last 10 min in scheme 1, while in line 421 the authors report 5 min, and 3 min in line 435.
- The sample trial was mentioned to last 5 min in scheme 1 and 3 min in line 440.
I strongly suggest that the authors refine their POE protocol in order to validate the that this paradigm can be used for the study of emotional modulation of memory consolidation.
Comments on the Quality of English LanguageI recommend the authors to correct some typos and to perform moderate editing to English language.
Author Response
Report #1
Thank you for the careful evaluation of our manuscript. We have revised the manuscript taking into account the suggested modifications. All changes in the MS are highlighted.
The authors aim to study the effect of endogenous NA release in the BLA induced by an aversive stimulus on memory consolidation for a neutral hippocampal-dependent task.
For that they used the behavioral paradigm object location task (OLT) combined with mild predator odor exposure (POE). At the same time the authors aimed at showing that this paradigm can be used for the study of emotional modulation of memory consolidation.
Point #1: I found the title of the article very confusing and the behavioral results obtained are not really in agreement with this title.
Response: We agree with the reviewer's comment. The title has been changed. Now it sounds like:
“ Predatory odor exposure as a potential paradigm for studying emotional modulation of memory consolidation - the role of the noradrenergic transmission in the basolateral amygdala”
Point #2: Also, the introduction is very long. The authors do not need to give that many details in this section.
Response: Following the Reviewer's recommendation, the Introduction was shortened. We deleted some sentences that give details not strictly related to the paradigm used in our study.
Point #3: The authors predicted that memory enhancement prompt by POE may be dependent on NA transmission in BLA. However, the results of this article do not support this.
- First, the results do not show that POE is capable of inducing a strong effect on memory consolidation for a weaker behavioral task (Fig. 3)
- Additionally, if the authors would also need to measure/confirm the release of NA in the BLA during their paradigm to full state that the stress induced by POE is dependent on NA transmission in BLA.
Response: Thank you for your interest in our work! Indeed, the results of the current study do not show a substantial effect on the performance of the animals in the test session. In the Discussion section of our manuscript, we have outlined several reasons for this lack of effect. Please note that, as stated in the manuscript, this is a pilot study and several features of the experimental design need to be carefully re-evaluated. We plan to optimize the protocol shortly and obtain more convincing evidence of the modulatory effects of POE on initial memory consolidation. At present, many of the factors involved in the effects of acute stress on hippocampal-dependent memory require further elucidation. Nevertheless, the current study has provided some evidence that POE can induce changes in the expression of the plasticity-related markers of interest.
Moreover, the pharmacological treatment we used suggests that the variation in the expression of both pCREB and Arc in the hippocampus is influenced by the BLA, specifically by the noradrenergic transmission there. While the focus of the current study was on noradrenaline, another neuromodulator (e.g. acetylcholine) and corticosterone in the BLA could also influence experience-dependent plasticity in the hippocampus. We agree that methods such as fiber photometry and in vivo microdialysis will allow us to understand the role of norepinephrine better. Our protocol will be of interest to other researchers who can perform this type of experiment.
Point #4: There are some discrepancies across the manuscript in what concerns the experimental procedures between scheme 1 and the text:
- Habituation to the OF box was mentioned to last 10 min in scheme 1, while in line 421 the authors report 5 min, and 3 min in line 435.
- The sample trial was mentioned to last 5 min in scheme 1 and 3 min in line 440.
Response: Figures 1 and 2 (two schemes) were corrected - the names of the groups and which groups were exposed to petri (empty or with urine). Also the information on the duration of each step during the sampling and testing phase, the intervals between different procedures were carefully checked again and if there was any technical error it was corrected.
Point #5: I strongly suggest that the authors refine their POE protocol in order to validate that this paradigm can be used for the study of emotional modulation of memory consolidation.
Response: Your constructive feedback is greatly appreciated! As stated in the manuscript, this is a pilot study and we are aware that several issues need to be carefully evaluated. We agree that tracking norepinephrine levels in the dorsal hippocampus and basolateral amygdala during POE will provide valuable insight into potential effects on memory consolidation and the effects of emotional arousal on memory quality. However, our laboratory currently lacks the equipment to conduct an experiment that would provide such data. In the Discussion section of the current manuscript, we have outlined several possibilities as to why we did not observe any behavioral effects during the test session, most notably the habituation of the animals prior to the start of the experiment and the short interval between the probe phase and the POE. We plan to run a new set of experiments soon to minimize the discrepancies and test different possibilities. Perhaps the weak versions of other behavioral protocols, such as contextual threat conditioning or inhibitory avoidance, will be more suitable for coupling with POE. Although our POE protocol is far from absolute, our current publication aims to stimulate future experiments that take advantage of more ethologically valid stimuli to investigate emotional modulation of memory consolidation, reorganization, and transformation. We believe that predator odors and other social olfactory stimuli have great potential and may elicit the expression of more diverse defensive behavioral responses. In addition, we hope that our work will attract the interest of other research groups in the field who could further develop, modify and improve the POE protocol.
Point #6: Comments on the Quality of English Language
I recommend the authors to correct some typos and to perform moderate editing to English language.
Response: We are grateful for the reviewer's comments. We have corrected all spelling mistakes and made the necessary edits to the new version of the manuscript.

Reviewer 2 Report
Comments and Suggestions for Authors
The manuscript entitled "Noradrenergic transmission in the basolateral amygdala mediates the effects of predatory odor exposure on hippocampal plasticity - a new concept for studying the emotional modulation of memory consolidation" by Peshev et al. investigates noradrenergic-dependent modulation of emotional memory.
To determine a role of endogenously released NA by the locus coeruleus into the BLA in hippocampus-dependent learning tasks, the authors designed two different experiments investigating 1) object location task sampling combined with predator exposure and subsequent analysis of pCREB levels by ELISA and Arc levels by Western blotting and 2) object location task sampling combined with predator exposure (POE) and subsequent object learning test and analysis of pCREB and Arc levels. To test a possible involvement of beta-adrenergic receptors, the authors infused the BLA of one group of animals with the non-selective beta-blocker propranolol.
The authors found that 4 minutes of predator exposure (coyote urine) caused a significant increase in pCREB levels compared to either control animals or animals that had been exposed to predator odor and subjected to propranolol treatment. In addition, a tendency towards increased Arc protein levels in the POE+veh group was detected. Despite increased pCREB levels in the POE+veh group, the authors did not detect any differences in the object learning test 24 hours after POE. Similarly, pCREB levels and Arc levels were similar in the 3 test groups, suggesting lack of evidence for a POE-induced modulatory influence of memory consolidation.
The authors clearly describe their study as a pilot study and discuss potential sources that might have influenced lack of evidence. The most important probably being the short interval between object learning sampling and odor exposure that might have induced a strong association between the 2 stimuli.
While the experimental design of the study seems solid, I would like to list a few concerns that should be addressed:
1) The authors state that ‘young’ male Wistar rats were used to experiments. Could the authors please specify the exact age of animals and could a peripubertal age possibly have influenced the outcome of the study?
2) Have the authors tested whether POE increased corticosterone levels in their experimental design?
3) I would also recommend placing an empty petridish in the control group (C+veh) environment.
4) The authors chose to expose the animals to predator odor for 4 minutes. Any rationale for this duration? Could the duration of odor exposure have influenced memory consolidation? Please also note that odor exposure duration in materials and methods is given as 5 min in contrast to Scheme 1.
5) Have the authors observed aversive behavior towards predator odor during exposure?
6) The authors state that pCREB and/or Arc protein levels were detected in hippocampal samples. Have the authors used whole hippocampal tissue for Arc and/or pCREB detection? As the authors elaborate, the dorsal and ventral hippocampus are in many aspects functionally distinct areas, especially with regard to memory and emotion. If homogenates of whole hippocampi were used in this study, this should be addressed and discussed. In future, it might be a good idea to sample dorsal and ventral hippocampi independently.
7) p5, ll. 205-208, the authors state that they investigated “the role of endogenous NA released from the BLA as a result of POE on the expression of plasticity marker (pCREB) in the hippocampus, …. Is there any evidence that predator exposure increases NA release from NAergic neurons in the BLA and what is the percentage of NAergic neurons in the BLA? Did the authors possibly mean LC-induced NA release into the BLA?
8) Scheme 1 is misleading as it implies that there are 3 groups of animals (control, stress+POE, stress+prop) BEFORE the beginning of the experiment and that all 3 groups receive coyote urine during the actual experiment. This scheme should be changed accordingly. In addition, the names of the 3 groups are different in the text (C+veh, POE+veh, POE+prop).
9) Fig 1: According to Materials and Methods and Scheme 1, 7 animals were used for experiment 1, however, Fig. 1 POE + veh group is short of 2 animals. Could the authors please specify why to excluded 2 animals from the analysis?
Fig 2 contains data from N=6 animals. Please explain the discrepancy between Scheme 1 and Fig. 2 animal numbers.
Fig 4: Please check animal numbers. Discrepancy between Scheme 1 and Fig. 4
Fig. 5: Please check n number provided in figure legend with bar chart. In addition, discrepancy between Scheme 1 and Fig. 5.
Some editorial aspects:
P1, line 38 „Episodic memories requires“. Please note that this sentence should read “epidosic memories require”
p. 3, ll. 100-101: please rephrase this sentence.
p. 3, l. 105 “heighten corticosterone level”: this should read “heightened corticosterone levels”
p. 3, l 124: “system injection of corticosterone”; possibly: systemic injection?
p. 3, l. 142 “inducted by learning”, possibly induced by learning?
p. 3, l. 147: “ORT task”; this should read ORT as abbreviation ORT has been introduced as object recognition task.
p. 4, ll. 178-179: “memory consolidation prompt by POE”; possibly prompted by?
p. 6, l. 212 Typo propanonol
The introduction and discussion are too long and read like a review at times. The authors should try to shorten both and focus on BLA-induced changes in memory modulation/consolidation.
Comments on the Quality of English Language
The manuscript is written in good English, however, I recommend its editing of English language.
Author Response
Report #1
Point #1: While the experimental design of the study seems solid, I would like to list a few concerns that should be addressed:
1) The authors state that ‘young’ male Wistar rats were used to experiments. Could the authors please specify the exact age of animals and could a peripubertal age possibly have influenced the outcome of the study?
Response: The rats were mature – 60-day-old rats (young adult). We replaced “young adult” by “sixty-day-old rats”.
Point #2: 2) Have the authors tested whether POE increased corticosterone levels in their experimental design?
Response: In the current study, we have not yet considered obtaining data on corticosterone levels. Animals were subjected to infusion into the BLA of vehicle or propranolol. As this invasive procedure is very stressful in itself, we considered the possibility that taking blood samples from the tail vein might further increase the animals' anxiety levels after the experiment and affect their performance the next day. We wanted to limit any conditioning effect, where the experimental context is associated with potentially harmful outcomes that could lead to a lack of exploration. We could have taken blood samples from the animals in Experiment 1. However, increasing the intensity of the stress during this procedure could have had a negative effect on the levels of the molecular markers of interest.
Point #3: 3) I would also recommend placing an empty petridish in the control group (C+veh) environment.
Response: We intentionally did not apply this step to the C+veh group because it may induce a "novelty" effect in this group as well. Most behavioral tagging experiments report that novelty positively modulates memory consolidation. As the Petri dish was placed in a separate empty cage to which the animals had not been exposed, we considered the possibility that the novel environment might act as a minor stressor and affect performance in the subsequent test. However, it would be a good idea for our future experiments to habituate the animals to the cage before exposing them to the predator odor. This would eliminate novelty effects and allow us to better assess the modulatory potential of the aversive olfactory stimulus. We also considered using more than one petri dish to increase the intensity of the odor-induced stress response. In the current experiment, the petri dish contained a gauze infused with 7 mL of coyote urine obtained from Maine Outdoor Solutions, LLC.
Point #4: 4) The authors chose to expose the animals to predator odor for 4 minutes. Any rationale for this duration? Could the duration of odor exposure have influenced memory consolidation? Please also note that odor exposure duration in materials and methods is given as 5 min in contrast to Scheme 1.
Response: Our decision to expose animals to predator odor for 4 minutes was based on the findings of Migliaro et al (2021). In their study, they observed that exposure of Wistar rats to bobcat urine for 10 or 20 minutes increased anxiety and depression-like behavior 24 hours later, whereas exposure for 3 minutes did not have the same behavioral effect. We have described their work in the Discussion section of our manuscript. We chose this duration for the POE because we wanted our protocol to represent mildly stressful experiences, as high levels of stress usually have a negative effect on animal performance or increase anxiety-related behavious. This duration was also chosen based on the effects described by the "Yerkes-Dodson law" for optimal arousal, where mild to moderate levels of arousal may enhance performance on low-arousal (easy) tasks, but impair learning on more difficult/intense tasks. These effects are well described in a review by Diamond et al. (2007).
Figures 1 and 2 (two schemes) were corrected - the names of the groups and which groups were exposed to petri (empty or with urine). Also the information on the duration of each step during the sampling and testing phase, the intervals between different procedures were carefully checked again and if there was any technical error it was corrected.
Point #5: Have the authors observed aversive behavior towards predator odor during exposure?
Response: We have observed several different defensive or anxiety-related behaviors. Animals tend to spend more time on the opposite side of the cage and avoid approaching the Petri dish after initial exploration. We also observed risk-assessment behavior, manifested by stretching and sniffing from a distance. In addition, some rats showed significantly increased self-grooming. However, as our experiments were designed to investigate the effects of POE on memory consolidation and subsequent performance in the OL test, we did not further analyze the observed behaviors. Nevertheless, our observations are consistent with the repertoire of behaviors described in the article by Maestas-Olguin, Parish and Pentkowski (2021), who investigated the behavioral effects of coyote urine exposure in more detail. We have cited their work in the Introduction section of our manuscript.
Point # 6: 6) The authors state that pCREB and/or Arc protein levels were detected in hippocampal samples. Have the authors used whole hippocampal tissue for Arc and/or pCREB detection? As the authors elaborate, the dorsal and ventral hippocampus are in many aspects functionally distinct areas, especially with regard to memory and emotion. If homogenates of whole hippocampi were used in this study, this should be addressed and discussed. In future, it might be a good idea to sample dorsal and ventral hippocampi independently.
Response: Whole hippocampus was used for the biochemical assay. We fully agree with the reviewer’s recommendation to use a technique that allow samples from the dorsal and ventral hippocampus to be examined independently. In the discussion section of the manuscript (page: 11; lines; 330-335), we have already addressed the limitation of our approach.
Point # 7: 7) p5, ll. 205-208, the authors state that they investigated “the role of endogenous NA released from the BLA as a result of POE on the expression of plasticity marker (pCREB) in the hippocampus, …. Is there any evidence that predator exposure increases NA release from NAergic neurons in the BLA and what is the percentage of NAergic neurons in the BLA? Did the authors possibly mean LC-induced NA release into the BLA?
Response: Thank you for your detailed examination of the text! It is indeed a technical error and will be corrected. We believe that if our POE protocol is able to increase norepinephrine levels in the basolateral amygdala, this is due to increased activation of norepinephrine neuronal populations in the LC. Further experiments are needed to confirm or refute this hypothesis.
Point # 8: 8) Scheme 1 is misleading as it implies that there are 3 groups of animals (control, stress+POE, stress+prop) BEFORE the beginning of the experiment and that all 3 groups receive coyote urine during the actual experiment. This scheme should be changed accordingly. In addition, the names of the 3 groups are different in the text (C+veh, POE+veh, POE+prop).
Response: We are grateful for this important comment! All discrepancies between the information given in the first two figures and in the Methods have been corrected and are given with the track changes.
Point # 9: 9) Fig 1: According to Materials and Methods and Scheme 1, 7 animals were used for experiment 1, however, Fig. 1 POE + veh group is short of 2 animals. Could the authors please specify why to excluded 2 animals from the analysis?
Response: For experiment #1, n = 7 rats were used per group. At the end of the experiments, the hippocampi were distributed for analysis by ELISA (n = 5-7) or WB (n = 6) per group. The Scheme 1 (Experiment #1 and Experiment #2) was corrected. In each test (memory, WB or ELISA) the exact number of animals used was reported for clarity.
Point # 10: Fig 2 contains data from N=6 animals. Please explain the discrepancy between Scheme 1 and Fig. 2 animal numbers.
Response: Response to Point # 10.
Point # 11: Fig 4: Please check animal numbers. Discrepancy between Scheme 1 and Fig. 4 Fig. 5: Please check n number provided in figure legend with bar chart. In addition, discrepancy between Scheme 1 and Fig. 5.
Response: Corrected (the response to point # 9 and 11)
Point # 12: Some editorial aspects:
P1, line 38 „Episodic memories requires“. Please note that this sentence should read “epidosic memories require”
Response: Corrected.
Point # 13: p. 3, ll. 100-101: please rephrase this sentence.
Response: We are thankful for this note. The sentence was removed
Point # 14: p. 3, l 124: “system injection of corticosterone”; possibly: systemic injection?
Response: Corrected.
Point # 15: p. 3, l. 142 “inducted by learning”, possibly induced by learning?
Response: Corrected.
Point # 16: p. 3, l. 147: “ORT task”; this should read ORT as abbreviation ORT has been introduced as object recognition task.
Response: Corrected.
Point # 17: p. 4, ll. 178-179: “memory consolidation prompt by POE”; possibly prompted by?
Response: Corrected.
Point # 18 p. 6, l. 212 Typo propanonol
Response: Corrected.
Point # 19 The introduction and discussion are too long and read like a review at times. The authors should try to shorten both and focus on BLA-induced changes in memory modulation/consolidation.
Response: We're grateful for that recommendation. The introduction and discussion have been thoroughly edited and all changes are listed with track changes.
Point # 20 Comments on the Quality of English Language
The manuscript is written in good English, however, I recommend its editing of English language.
Response: We are grateful for the reviewer's comments. We have corrected all spelling mistakes and made the necessary edits to the new version of the manuscript.
Report #1
Point #1: While the experimental design of the study seems solid, I would like to list a few concerns that should be addressed:
1) The authors state that ‘young’ male Wistar rats were used to experiments. Could the authors please specify the exact age of animals and could a peripubertal age possibly have influenced the outcome of the study?
Response: The rats were mature – 60-day-old rats (young adult). We replaced “young adult” by “sixty-day-old rats”.
Point #2: 2) Have the authors tested whether POE increased corticosterone levels in their experimental design?
Response: In the current study, we have not yet considered obtaining data on corticosterone levels. Animals were subjected to infusion into the BLA of vehicle or propranolol. As this invasive procedure is very stressful in itself, we considered the possibility that taking blood samples from the tail vein might further increase the animals' anxiety levels after the experiment and affect their performance the next day. We wanted to limit any conditioning effect, where the experimental context is associated with potentially harmful outcomes that could lead to a lack of exploration. We could have taken blood samples from the animals in Experiment 1. However, increasing the intensity of the stress during this procedure could have had a negative effect on the levels of the molecular markers of interest.
Point #3: 3) I would also recommend placing an empty petridish in the control group (C+veh) environment.
Response: We intentionally did not apply this step to the C+veh group because it may induce a "novelty" effect in this group as well. Most behavioral tagging experiments report that novelty positively modulates memory consolidation. As the Petri dish was placed in a separate empty cage to which the animals had not been exposed, we considered the possibility that the novel environment might act as a minor stressor and affect performance in the subsequent test. However, it would be a good idea for our future experiments to habituate the animals to the cage before exposing them to the predator odor. This would eliminate novelty effects and allow us to better assess the modulatory potential of the aversive olfactory stimulus. We also considered using more than one petri dish to increase the intensity of the odor-induced stress response. In the current experiment, the petri dish contained a gauze infused with 7 mL of coyote urine obtained from Maine Outdoor Solutions, LLC.
Point #4: 4) The authors chose to expose the animals to predator odor for 4 minutes. Any rationale for this duration? Could the duration of odor exposure have influenced memory consolidation? Please also note that odor exposure duration in materials and methods is given as 5 min in contrast to Scheme 1.
Response: Our decision to expose animals to predator odor for 4 minutes was based on the findings of Migliaro et al (2021). In their study, they observed that exposure of Wistar rats to bobcat urine for 10 or 20 minutes increased anxiety and depression-like behavior 24 hours later, whereas exposure for 3 minutes did not have the same behavioral effect. We have described their work in the Discussion section of our manuscript. We chose this duration for the POE because we wanted our protocol to represent mildly stressful experiences, as high levels of stress usually have a negative effect on animal performance or increase anxiety-related behavious. This duration was also chosen based on the effects described by the "Yerkes-Dodson law" for optimal arousal, where mild to moderate levels of arousal may enhance performance on low-arousal (easy) tasks, but impair learning on more difficult/intense tasks. These effects are well described in a review by Diamond et al. (2007).
Figures 1 and 2 (two schemes) were corrected - the names of the groups and which groups were exposed to petri (empty or with urine). Also the information on the duration of each step during the sampling and testing phase, the intervals between different procedures were carefully checked again and if there was any technical error it was corrected.
Point #5: Have the authors observed aversive behavior towards predator odor during exposure?
Response: We have observed several different defensive or anxiety-related behaviors. Animals tend to spend more time on the opposite side of the cage and avoid approaching the Petri dish after initial exploration. We also observed risk-assessment behavior, manifested by stretching and sniffing from a distance. In addition, some rats showed significantly increased self-grooming. However, as our experiments were designed to investigate the effects of POE on memory consolidation and subsequent performance in the OL test, we did not further analyze the observed behaviors. Nevertheless, our observations are consistent with the repertoire of behaviors described in the article by Maestas-Olguin, Parish and Pentkowski (2021), who investigated the behavioral effects of coyote urine exposure in more detail. We have cited their work in the Introduction section of our manuscript.
Point # 6: 6) The authors state that pCREB and/or Arc protein levels were detected in hippocampal samples. Have the authors used whole hippocampal tissue for Arc and/or pCREB detection? As the authors elaborate, the dorsal and ventral hippocampus are in many aspects functionally distinct areas, especially with regard to memory and emotion. If homogenates of whole hippocampi were used in this study, this should be addressed and discussed. In future, it might be a good idea to sample dorsal and ventral hippocampi independently.
Response: Whole hippocampus was used for the biochemical assay. We fully agree with the reviewer’s recommendation to use a technique that allow samples from the dorsal and ventral hippocampus to be examined independently. In the discussion section of the manuscript (page: 11; lines; 330-335), we have already addressed the limitation of our approach.
Point # 7: 7) p5, ll. 205-208, the authors state that they investigated “the role of endogenous NA released from the BLA as a result of POE on the expression of plasticity marker (pCREB) in the hippocampus, …. Is there any evidence that predator exposure increases NA release from NAergic neurons in the BLA and what is the percentage of NAergic neurons in the BLA? Did the authors possibly mean LC-induced NA release into the BLA?
Response: Thank you for your detailed examination of the text! It is indeed a technical error and will be corrected. We believe that if our POE protocol is able to increase norepinephrine levels in the basolateral amygdala, this is due to increased activation of norepinephrine neuronal populations in the LC. Further experiments are needed to confirm or refute this hypothesis.
Point # 8: 8) Scheme 1 is misleading as it implies that there are 3 groups of animals (control, stress+POE, stress+prop) BEFORE the beginning of the experiment and that all 3 groups receive coyote urine during the actual experiment. This scheme should be changed accordingly. In addition, the names of the 3 groups are different in the text (C+veh, POE+veh, POE+prop).
Response: We are grateful for this important comment! All discrepancies between the information given in the first two figures and in the Methods have been corrected and are given with the track changes.
Point # 9: 9) Fig 1: According to Materials and Methods and Scheme 1, 7 animals were used for experiment 1, however, Fig. 1 POE + veh group is short of 2 animals. Could the authors please specify why to excluded 2 animals from the analysis?
Response: For experiment #1, n = 7 rats were used per group. At the end of the experiments, the hippocampi were distributed for analysis by ELISA (n = 5-7) or WB (n = 6) per group. The Scheme 1 (Experiment #1 and Experiment #2) was corrected. In each test (memory, WB or ELISA) the exact number of animals used was reported for clarity.
Point # 10: Fig 2 contains data from N=6 animals. Please explain the discrepancy between Scheme 1 and Fig. 2 animal numbers.
Response: Response to Point # 10.
Point # 11: Fig 4: Please check animal numbers. Discrepancy between Scheme 1 and Fig. 4 Fig. 5: Please check n number provided in figure legend with bar chart. In addition, discrepancy between Scheme 1 and Fig. 5.
Response: Corrected (the response to point # 9 and 11)
Point # 12: Some editorial aspects:
P1, line 38 „Episodic memories requires“. Please note that this sentence should read “epidosic memories require”
Response: Corrected.
Point # 13: p. 3, ll. 100-101: please rephrase this sentence.
Response: We are thankful for this note. The sentence was removed
Point # 14: p. 3, l 124: “system injection of corticosterone”; possibly: systemic injection?
Response: Corrected.
Point # 15: p. 3, l. 142 “inducted by learning”, possibly induced by learning?
Response: Corrected.
Point # 16: p. 3, l. 147: “ORT task”; this should read ORT as abbreviation ORT has been introduced as object recognition task.
Response: Corrected.
Point # 17: p. 4, ll. 178-179: “memory consolidation prompt by POE”; possibly prompted by?
Response: Corrected.
Point # 18 p. 6, l. 212 Typo propanonol
Response: Corrected.
Point # 19 The introduction and discussion are too long and read like a review at times. The authors should try to shorten both and focus on BLA-induced changes in memory modulation/consolidation.
Response: We're grateful for that recommendation. The introduction and discussion have been thoroughly edited and all changes are listed with track changes.
Point # 20 Comments on the Quality of English Language
The manuscript is written in good English, however, I recommend its editing of English language.
Response: We are grateful for the reviewer's comments. We have corrected all spelling mistakes and made the necessary edits to the new version of the manuscript.
Report #1
Point #1: While the experimental design of the study seems solid, I would like to list a few concerns that should be addressed:
1) The authors state that ‘young’ male Wistar rats were used to experiments. Could the authors please specify the exact age of animals and could a peripubertal age possibly have influenced the outcome of the study?
Response: The rats were mature – 60-day-old rats (young adult). We replaced “young adult” by “sixty-day-old rats”.
Point #2: 2) Have the authors tested whether POE increased corticosterone levels in their experimental design?
Response: In the current study, we have not yet considered obtaining data on corticosterone levels. Animals were subjected to infusion into the BLA of vehicle or propranolol. As this invasive procedure is very stressful in itself, we considered the possibility that taking blood samples from the tail vein might further increase the animals' anxiety levels after the experiment and affect their performance the next day. We wanted to limit any conditioning effect, where the experimental context is associated with potentially harmful outcomes that could lead to a lack of exploration. We could have taken blood samples from the animals in Experiment 1. However, increasing the intensity of the stress during this procedure could have had a negative effect on the levels of the molecular markers of interest.
Point #3: 3) I would also recommend placing an empty petridish in the control group (C+veh) environment.
Response: We intentionally did not apply this step to the C+veh group because it may induce a "novelty" effect in this group as well. Most behavioral tagging experiments report that novelty positively modulates memory consolidation. As the Petri dish was placed in a separate empty cage to which the animals had not been exposed, we considered the possibility that the novel environment might act as a minor stressor and affect performance in the subsequent test. However, it would be a good idea for our future experiments to habituate the animals to the cage before exposing them to the predator odor. This would eliminate novelty effects and allow us to better assess the modulatory potential of the aversive olfactory stimulus. We also considered using more than one petri dish to increase the intensity of the odor-induced stress response. In the current experiment, the petri dish contained a gauze infused with 7 mL of coyote urine obtained from Maine Outdoor Solutions, LLC.
Point #4: 4) The authors chose to expose the animals to predator odor for 4 minutes. Any rationale for this duration? Could the duration of odor exposure have influenced memory consolidation? Please also note that odor exposure duration in materials and methods is given as 5 min in contrast to Scheme 1.
Response: Our decision to expose animals to predator odor for 4 minutes was based on the findings of Migliaro et al (2021). In their study, they observed that exposure of Wistar rats to bobcat urine for 10 or 20 minutes increased anxiety and depression-like behavior 24 hours later, whereas exposure for 3 minutes did not have the same behavioral effect. We have described their work in the Discussion section of our manuscript. We chose this duration for the POE because we wanted our protocol to represent mildly stressful experiences, as high levels of stress usually have a negative effect on animal performance or increase anxiety-related behavious. This duration was also chosen based on the effects described by the "Yerkes-Dodson law" for optimal arousal, where mild to moderate levels of arousal may enhance performance on low-arousal (easy) tasks, but impair learning on more difficult/intense tasks. These effects are well described in a review by Diamond et al. (2007).
Figures 1 and 2 (two schemes) were corrected - the names of the groups and which groups were exposed to petri (empty or with urine). Also the information on the duration of each step during the sampling and testing phase, the intervals between different procedures were carefully checked again and if there was any technical error it was corrected.
Point #5: Have the authors observed aversive behavior towards predator odor during exposure?
Response: We have observed several different defensive or anxiety-related behaviors. Animals tend to spend more time on the opposite side of the cage and avoid approaching the Petri dish after initial exploration. We also observed risk-assessment behavior, manifested by stretching and sniffing from a distance. In addition, some rats showed significantly increased self-grooming. However, as our experiments were designed to investigate the effects of POE on memory consolidation and subsequent performance in the OL test, we did not further analyze the observed behaviors. Nevertheless, our observations are consistent with the repertoire of behaviors described in the article by Maestas-Olguin, Parish and Pentkowski (2021), who investigated the behavioral effects of coyote urine exposure in more detail. We have cited their work in the Introduction section of our manuscript.
Point # 6: 6) The authors state that pCREB and/or Arc protein levels were detected in hippocampal samples. Have the authors used whole hippocampal tissue for Arc and/or pCREB detection? As the authors elaborate, the dorsal and ventral hippocampus are in many aspects functionally distinct areas, especially with regard to memory and emotion. If homogenates of whole hippocampi were used in this study, this should be addressed and discussed. In future, it might be a good idea to sample dorsal and ventral hippocampi independently.
Response: Whole hippocampus was used for the biochemical assay. We fully agree with the reviewer’s recommendation to use a technique that allow samples from the dorsal and ventral hippocampus to be examined independently. In the discussion section of the manuscript (page: 11; lines; 330-335), we have already addressed the limitation of our approach.
Point # 7: 7) p5, ll. 205-208, the authors state that they investigated “the role of endogenous NA released from the BLA as a result of POE on the expression of plasticity marker (pCREB) in the hippocampus, …. Is there any evidence that predator exposure increases NA release from NAergic neurons in the BLA and what is the percentage of NAergic neurons in the BLA? Did the authors possibly mean LC-induced NA release into the BLA?
Response: Thank you for your detailed examination of the text! It is indeed a technical error and will be corrected. We believe that if our POE protocol is able to increase norepinephrine levels in the basolateral amygdala, this is due to increased activation of norepinephrine neuronal populations in the LC. Further experiments are needed to confirm or refute this hypothesis.
Point # 8: 8) Scheme 1 is misleading as it implies that there are 3 groups of animals (control, stress+POE, stress+prop) BEFORE the beginning of the experiment and that all 3 groups receive coyote urine during the actual experiment. This scheme should be changed accordingly. In addition, the names of the 3 groups are different in the text (C+veh, POE+veh, POE+prop).
Response: We are grateful for this important comment! All discrepancies between the information given in the first two figures and in the Methods have been corrected and are given with the track changes.
Point # 9: 9) Fig 1: According to Materials and Methods and Scheme 1, 7 animals were used for experiment 1, however, Fig. 1 POE + veh group is short of 2 animals. Could the authors please specify why to excluded 2 animals from the analysis?
Response: For experiment #1, n = 7 rats were used per group. At the end of the experiments, the hippocampi were distributed for analysis by ELISA (n = 5-7) or WB (n = 6) per group. The Scheme 1 (Experiment #1 and Experiment #2) was corrected. In each test (memory, WB or ELISA) the exact number of animals used was reported for clarity.
Point # 10: Fig 2 contains data from N=6 animals. Please explain the discrepancy between Scheme 1 and Fig. 2 animal numbers.
Response: Response to Point # 10.
Point # 11: Fig 4: Please check animal numbers. Discrepancy between Scheme 1 and Fig. 4 Fig. 5: Please check n number provided in figure legend with bar chart. In addition, discrepancy between Scheme 1 and Fig. 5.
Response: Corrected (the response to point # 9 and 11)
Point # 12: Some editorial aspects:
P1, line 38 „Episodic memories requires“. Please note that this sentence should read “epidosic memories require”
Response: Corrected.
Point # 13: p. 3, ll. 100-101: please rephrase this sentence.
Response: We are thankful for this note. The sentence was removed
Point # 14: p. 3, l 124: “system injection of corticosterone”; possibly: systemic injection?
Response: Corrected.
Point # 15: p. 3, l. 142 “inducted by learning”, possibly induced by learning?
Response: Corrected.
Point # 16: p. 3, l. 147: “ORT task”; this should read ORT as abbreviation ORT has been introduced as object recognition task.
Response: Corrected.
Point # 17: p. 4, ll. 178-179: “memory consolidation prompt by POE”; possibly prompted by?
Response: Corrected.
Point # 18 p. 6, l. 212 Typo propanonol
Response: Corrected.
Point # 19 The introduction and discussion are too long and read like a review at times. The authors should try to shorten both and focus on BLA-induced changes in memory modulation/consolidation.
Response: We're grateful for that recommendation. The introduction and discussion have been thoroughly edited and all changes are listed with track changes.
Point # 20 Comments on the Quality of English Language
The manuscript is written in good English, however, I recommend its editing of English language.
Response: We are grateful for the reviewer's comments. We have corrected all spelling mistakes and made the necessary edits to the new version of the manuscript.
Report #1
Point #1: While the experimental design of the study seems solid, I would like to list a few concerns that should be addressed:
1) The authors state that ‘young’ male Wistar rats were used to experiments. Could the authors please specify the exact age of animals and could a peripubertal age possibly have influenced the outcome of the study?
Response: The rats were mature – 60-day-old rats (young adult). We replaced “young adult” by “sixty-day-old rats”.
Point #2: 2) Have the authors tested whether POE increased corticosterone levels in their experimental design?
Response: In the current study, we have not yet considered obtaining data on corticosterone levels. Animals were subjected to infusion into the BLA of vehicle or propranolol. As this invasive procedure is very stressful in itself, we considered the possibility that taking blood samples from the tail vein might further increase the animals' anxiety levels after the experiment and affect their performance the next day. We wanted to limit any conditioning effect, where the experimental context is associated with potentially harmful outcomes that could lead to a lack of exploration. We could have taken blood samples from the animals in Experiment 1. However, increasing the intensity of the stress during this procedure could have had a negative effect on the levels of the molecular markers of interest.
Point #3: 3) I would also recommend placing an empty petridish in the control group (C+veh) environment.
Response: We intentionally did not apply this step to the C+veh group because it may induce a "novelty" effect in this group as well. Most behavioral tagging experiments report that novelty positively modulates memory consolidation. As the Petri dish was placed in a separate empty cage to which the animals had not been exposed, we considered the possibility that the novel environment might act as a minor stressor and affect performance in the subsequent test. However, it would be a good idea for our future experiments to habituate the animals to the cage before exposing them to the predator odor. This would eliminate novelty effects and allow us to better assess the modulatory potential of the aversive olfactory stimulus. We also considered using more than one petri dish to increase the intensity of the odor-induced stress response. In the current experiment, the petri dish contained a gauze infused with 7 mL of coyote urine obtained from Maine Outdoor Solutions, LLC.
Point #4: 4) The authors chose to expose the animals to predator odor for 4 minutes. Any rationale for this duration? Could the duration of odor exposure have influenced memory consolidation? Please also note that odor exposure duration in materials and methods is given as 5 min in contrast to Scheme 1.
Response: Our decision to expose animals to predator odor for 4 minutes was based on the findings of Migliaro et al (2021). In their study, they observed that exposure of Wistar rats to bobcat urine for 10 or 20 minutes increased anxiety and depression-like behavior 24 hours later, whereas exposure for 3 minutes did not have the same behavioral effect. We have described their work in the Discussion section of our manuscript. We chose this duration for the POE because we wanted our protocol to represent mildly stressful experiences, as high levels of stress usually have a negative effect on animal performance or increase anxiety-related behavious. This duration was also chosen based on the effects described by the "Yerkes-Dodson law" for optimal arousal, where mild to moderate levels of arousal may enhance performance on low-arousal (easy) tasks, but impair learning on more difficult/intense tasks. These effects are well described in a review by Diamond et al. (2007).
Figures 1 and 2 (two schemes) were corrected - the names of the groups and which groups were exposed to petri (empty or with urine). Also the information on the duration of each step during the sampling and testing phase, the intervals between different procedures were carefully checked again and if there was any technical error it was corrected.
Point #5: Have the authors observed aversive behavior towards predator odor during exposure?
Response: We have observed several different defensive or anxiety-related behaviors. Animals tend to spend more time on the opposite side of the cage and avoid approaching the Petri dish after initial exploration. We also observed risk-assessment behavior, manifested by stretching and sniffing from a distance. In addition, some rats showed significantly increased self-grooming. However, as our experiments were designed to investigate the effects of POE on memory consolidation and subsequent performance in the OL test, we did not further analyze the observed behaviors. Nevertheless, our observations are consistent with the repertoire of behaviors described in the article by Maestas-Olguin, Parish and Pentkowski (2021), who investigated the behavioral effects of coyote urine exposure in more detail. We have cited their work in the Introduction section of our manuscript.
Point # 6: 6) The authors state that pCREB and/or Arc protein levels were detected in hippocampal samples. Have the authors used whole hippocampal tissue for Arc and/or pCREB detection? As the authors elaborate, the dorsal and ventral hippocampus are in many aspects functionally distinct areas, especially with regard to memory and emotion. If homogenates of whole hippocampi were used in this study, this should be addressed and discussed. In future, it might be a good idea to sample dorsal and ventral hippocampi independently.
Response: Whole hippocampus was used for the biochemical assay. We fully agree with the reviewer’s recommendation to use a technique that allow samples from the dorsal and ventral hippocampus to be examined independently. In the discussion section of the manuscript (page: 11; lines; 330-335), we have already addressed the limitation of our approach.
Point # 7: 7) p5, ll. 205-208, the authors state that they investigated “the role of endogenous NA released from the BLA as a result of POE on the expression of plasticity marker (pCREB) in the hippocampus, …. Is there any evidence that predator exposure increases NA release from NAergic neurons in the BLA and what is the percentage of NAergic neurons in the BLA? Did the authors possibly mean LC-induced NA release into the BLA?
Response: Thank you for your detailed examination of the text! It is indeed a technical error and will be corrected. We believe that if our POE protocol is able to increase norepinephrine levels in the basolateral amygdala, this is due to increased activation of norepinephrine neuronal populations in the LC. Further experiments are needed to confirm or refute this hypothesis.
Point # 8: 8) Scheme 1 is misleading as it implies that there are 3 groups of animals (control, stress+POE, stress+prop) BEFORE the beginning of the experiment and that all 3 groups receive coyote urine during the actual experiment. This scheme should be changed accordingly. In addition, the names of the 3 groups are different in the text (C+veh, POE+veh, POE+prop).
Response: We are grateful for this important comment! All discrepancies between the information given in the first two figures and in the Methods have been corrected and are given with the track changes.
Point # 9: 9) Fig 1: According to Materials and Methods and Scheme 1, 7 animals were used for experiment 1, however, Fig. 1 POE + veh group is short of 2 animals. Could the authors please specify why to excluded 2 animals from the analysis?
Response: For experiment #1, n = 7 rats were used per group. At the end of the experiments, the hippocampi were distributed for analysis by ELISA (n = 5-7) or WB (n = 6) per group. The Scheme 1 (Experiment #1 and Experiment #2) was corrected. In each test (memory, WB or ELISA) the exact number of animals used was reported for clarity.
Point # 10: Fig 2 contains data from N=6 animals. Please explain the discrepancy between Scheme 1 and Fig. 2 animal numbers.
Response: Response to Point # 10.
Point # 11: Fig 4: Please check animal numbers. Discrepancy between Scheme 1 and Fig. 4 Fig. 5: Please check n number provided in figure legend with bar chart. In addition, discrepancy between Scheme 1 and Fig. 5.
Response: Corrected (the response to point # 9 and 11)
Point # 12: Some editorial aspects:
P1, line 38 „Episodic memories requires“. Please note that this sentence should read “epidosic memories require”
Response: Corrected.
Point # 13: p. 3, ll. 100-101: please rephrase this sentence.
Response: We are thankful for this note. The sentence was removed
Point # 14: p. 3, l 124: “system injection of corticosterone”; possibly: systemic injection?
Response: Corrected.
Point # 15: p. 3, l. 142 “inducted by learning”, possibly induced by learning?
Response: Corrected.
Point # 16: p. 3, l. 147: “ORT task”; this should read ORT as abbreviation ORT has been introduced as object recognition task.
Response: Corrected.
Point # 17: p. 4, ll. 178-179: “memory consolidation prompt by POE”; possibly prompted by?
Response: Corrected.
Point # 18 p. 6, l. 212 Typo propanonol
Response: Corrected.
Point # 19 The introduction and discussion are too long and read like a review at times. The authors should try to shorten both and focus on BLA-induced changes in memory modulation/consolidation.
Response: We're grateful for that recommendation. The introduction and discussion have been thoroughly edited and all changes are listed with track changes.
Point # 20 Comments on the Quality of English Language
The manuscript is written in good English, however, I recommend its editing of English language.
Response: We are grateful for the reviewer's comments. We have corrected all spelling mistakes and made the necessary edits to the new version of the manuscript.
Report #1
Point #1: While the experimental design of the study seems solid, I would like to list a few concerns that should be addressed:
1) The authors state that ‘young’ male Wistar rats were used to experiments. Could the authors please specify the exact age of animals and could a peripubertal age possibly have influenced the outcome of the study?
Response: The rats were mature – 60-day-old rats (young adult). We replaced “young adult” by “sixty-day-old rats”.
Point #2: 2) Have the authors tested whether POE increased corticosterone levels in their experimental design?
Response: In the current study, we have not yet considered obtaining data on corticosterone levels. Animals were subjected to infusion into the BLA of vehicle or propranolol. As this invasive procedure is very stressful in itself, we considered the possibility that taking blood samples from the tail vein might further increase the animals' anxiety levels after the experiment and affect their performance the next day. We wanted to limit any conditioning effect, where the experimental context is associated with potentially harmful outcomes that could lead to a lack of exploration. We could have taken blood samples from the animals in Experiment 1. However, increasing the intensity of the stress during this procedure could have had a negative effect on the levels of the molecular markers of interest.
Point #3: 3) I would also recommend placing an empty petridish in the control group (C+veh) environment.
Response: We intentionally did not apply this step to the C+veh group because it may induce a "novelty" effect in this group as well. Most behavioral tagging experiments report that novelty positively modulates memory consolidation. As the Petri dish was placed in a separate empty cage to which the animals had not been exposed, we considered the possibility that the novel environment might act as a minor stressor and affect performance in the subsequent test. However, it would be a good idea for our future experiments to habituate the animals to the cage before exposing them to the predator odor. This would eliminate novelty effects and allow us to better assess the modulatory potential of the aversive olfactory stimulus. We also considered using more than one petri dish to increase the intensity of the odor-induced stress response. In the current experiment, the petri dish contained a gauze infused with 7 mL of coyote urine obtained from Maine Outdoor Solutions, LLC.
Point #4: 4) The authors chose to expose the animals to predator odor for 4 minutes. Any rationale for this duration? Could the duration of odor exposure have influenced memory consolidation? Please also note that odor exposure duration in materials and methods is given as 5 min in contrast to Scheme 1.
Response: Our decision to expose animals to predator odor for 4 minutes was based on the findings of Migliaro et al (2021). In their study, they observed that exposure of Wistar rats to bobcat urine for 10 or 20 minutes increased anxiety and depression-like behavior 24 hours later, whereas exposure for 3 minutes did not have the same behavioral effect. We have described their work in the Discussion section of our manuscript. We chose this duration for the POE because we wanted our protocol to represent mildly stressful experiences, as high levels of stress usually have a negative effect on animal performance or increase anxiety-related behavious. This duration was also chosen based on the effects described by the "Yerkes-Dodson law" for optimal arousal, where mild to moderate levels of arousal may enhance performance on low-arousal (easy) tasks, but impair learning on more difficult/intense tasks. These effects are well described in a review by Diamond et al. (2007).
Figures 1 and 2 (two schemes) were corrected - the names of the groups and which groups were exposed to petri (empty or with urine). Also the information on the duration of each step during the sampling and testing phase, the intervals between different procedures were carefully checked again and if there was any technical error it was corrected.
Point #5: Have the authors observed aversive behavior towards predator odor during exposure?
Response: We have observed several different defensive or anxiety-related behaviors. Animals tend to spend more time on the opposite side of the cage and avoid approaching the Petri dish after initial exploration. We also observed risk-assessment behavior, manifested by stretching and sniffing from a distance. In addition, some rats showed significantly increased self-grooming. However, as our experiments were designed to investigate the effects of POE on memory consolidation and subsequent performance in the OL test, we did not further analyze the observed behaviors. Nevertheless, our observations are consistent with the repertoire of behaviors described in the article by Maestas-Olguin, Parish and Pentkowski (2021), who investigated the behavioral effects of coyote urine exposure in more detail. We have cited their work in the Introduction section of our manuscript.
Point # 6: 6) The authors state that pCREB and/or Arc protein levels were detected in hippocampal samples. Have the authors used whole hippocampal tissue for Arc and/or pCREB detection? As the authors elaborate, the dorsal and ventral hippocampus are in many aspects functionally distinct areas, especially with regard to memory and emotion. If homogenates of whole hippocampi were used in this study, this should be addressed and discussed. In future, it might be a good idea to sample dorsal and ventral hippocampi independently.
Response: Whole hippocampus was used for the biochemical assay. We fully agree with the reviewer’s recommendation to use a technique that allow samples from the dorsal and ventral hippocampus to be examined independently. In the discussion section of the manuscript (page: 11; lines; 330-335), we have already addressed the limitation of our approach.
Point # 7: 7) p5, ll. 205-208, the authors state that they investigated “the role of endogenous NA released from the BLA as a result of POE on the expression of plasticity marker (pCREB) in the hippocampus, …. Is there any evidence that predator exposure increases NA release from NAergic neurons in the BLA and what is the percentage of NAergic neurons in the BLA? Did the authors possibly mean LC-induced NA release into the BLA?
Response: Thank you for your detailed examination of the text! It is indeed a technical error and will be corrected. We believe that if our POE protocol is able to increase norepinephrine levels in the basolateral amygdala, this is due to increased activation of norepinephrine neuronal populations in the LC. Further experiments are needed to confirm or refute this hypothesis.
Point # 8: 8) Scheme 1 is misleading as it implies that there are 3 groups of animals (control, stress+POE, stress+prop) BEFORE the beginning of the experiment and that all 3 groups receive coyote urine during the actual experiment. This scheme should be changed accordingly. In addition, the names of the 3 groups are different in the text (C+veh, POE+veh, POE+prop).
Response: We are grateful for this important comment! All discrepancies between the information given in the first two figures and in the Methods have been corrected and are given with the track changes.
Point # 9: 9) Fig 1: According to Materials and Methods and Scheme 1, 7 animals were used for experiment 1, however, Fig. 1 POE + veh group is short of 2 animals. Could the authors please specify why to excluded 2 animals from the analysis?
Response: For experiment #1, n = 7 rats were used per group. At the end of the experiments, the hippocampi were distributed for analysis by ELISA (n = 5-7) or WB (n = 6) per group. The Scheme 1 (Experiment #1 and Experiment #2) was corrected. In each test (memory, WB or ELISA) the exact number of animals used was reported for clarity.
Point # 10: Fig 2 contains data from N=6 animals. Please explain the discrepancy between Scheme 1 and Fig. 2 animal numbers.
Response: Response to Point # 10.
Point # 11: Fig 4: Please check animal numbers. Discrepancy between Scheme 1 and Fig. 4 Fig. 5: Please check n number provided in figure legend with bar chart. In addition, discrepancy between Scheme 1 and Fig. 5.
Response: Corrected (the response to point # 9 and 11)
Point # 12: Some editorial aspects:
P1, line 38 „Episodic memories requires“. Please note that this sentence should read “epidosic memories require”
Response: Corrected.
Point # 13: p. 3, ll. 100-101: please rephrase this sentence.
Response: We are thankful for this note. The sentence was removed
Point # 14: p. 3, l 124: “system injection of corticosterone”; possibly: systemic injection?
Response: Corrected.
Point # 15: p. 3, l. 142 “inducted by learning”, possibly induced by learning?
Response: Corrected.
Point # 16: p. 3, l. 147: “ORT task”; this should read ORT as abbreviation ORT has been introduced as object recognition task.
Response: Corrected.
Point # 17: p. 4, ll. 178-179: “memory consolidation prompt by POE”; possibly prompted by?
Response: Corrected.
Point # 18 p. 6, l. 212 Typo propanonol
Response: Corrected.
Point # 19 The introduction and discussion are too long and read like a review at times. The authors should try to shorten both and focus on BLA-induced changes in memory modulation/consolidation.
Response: We're grateful for that recommendation. The introduction and discussion have been thoroughly edited and all changes are listed with track changes.
Point # 20 Comments on the Quality of English Language
The manuscript is written in good English, however, I recommend its editing of English language.
Response: We are grateful for the reviewer's comments. We have corrected all spelling mistakes and made the necessary edits to the new version of the manuscript.
